# Does receiving a SARS-CoV-2 antibody test result change COVID-19 protective behaviors? Testing risk compensation in undergraduate students with a randomized controlled trial

Christina Ludema[1][☯], Molly S. Rosenberg[1][☯]*, Jonathan T. Macy[2], Sina Kianersi[1], Maya Luetke[1], Chen Chen[1], Lilian Golzarri-Arroyo[3], Erin Ables[3], Kevin Maki[2], David B. Allison[1]

1 Department of Epidemiology and Biostatistics, Indiana University School of Public Health-Bloomington, Bloomington, Indiana, 2 Department of Applied Health Science, Indiana University School of Public Health-Bloomington, Bloomington, Indiana, 3 Biostatistics Consulting Center, Department of Epidemiology and Biostatistics, Indiana University School of Public Health-Bloomington, Bloomington, Indiana

☯ These authors contributed equally to this work.
* rosenmol@indiana.edu

## Abstract

### Background

Risk compensation, or matching behavior to a perceived level of acceptable risk, can blunt the effectiveness of public health interventions. One area of possible risk compensation during the SARS-CoV-2 pandemic is antibody testing. While antibody tests are imperfect measures of immunity, results may influence risk perception and individual preventive actions. We conducted a randomized control trial to assess whether receiving antibody test results changed SARS-CoV-2 protective behaviors.

### Purpose

Assess whether objective information about antibody status, particularly for those who are antibody negative and likely still susceptible to SARS-CoV-2 infection, increases protective behaviors. Secondarily, assess whether a positive antibody test results in decreased protective behaviors.

### Methods

In September 2020, we enrolled 1076 undergraduate students, used fingerstick tests for SARS-CoV-2 antibodies, and randomized participants to receive their results immediately or delayed by 4 weeks. Two weeks later, participants completed a survey about their engagement in 4 protective behaviors (mask use, social event avoidance, staying home from work/school, ensuring physical distancing). We estimated differences between conditions for each of these behaviors, stratified by antibody status. For negative participants at baseline, we also estimated the difference between conditions for seroconversion over 8 weeks of follow-up.

**Data Availability Statement:** All relevant data are within the manuscript and its Supporting Information files.

**Funding:** This trial was supported by a charitable donation to the Indiana University Foundation, and antibody tests were donated by the United Arab Emirates. The funders had no role in study design, data collection and analysis, decision to publish, or preparation of the manuscript.

**Competing interests:** CL, MR, JT, SK, ML, CC, LGA, and EA have declared that no competing interests exist. We have read the journal's policy and KM has the following competing interests: 1. Consulting regarding development of therapeutics for Covid-19 (not related to the current manuscript) and payments made to a private clinic in which I am a partner for conduct of a Covid treatment trials (not related to the current manuscript) 2. Payments made to a private clinic in which I am a partner for conduct of a vaccine trial (not related to the current manuscript) DBA has the following competing interests: 1. Personal payments or promises for same from: Alkermes, Inc.; Biofortis Innovation Services (Merieux NutriSciences); Kaleido Biosciences; Medpace/Gelesis; and Novo Nordisk Fonden. 2. DBA's institution, Indiana University, and the Indiana University Foundation have received funds or donations to support his research or educational activities from: Dynamic Air Quality Solutions; Eli Lilly and Company; Pfizer; Reckitt Benckiser Group PLC; Soleno Therapeutics; United Arab Emirates Mission to the United Nations; and numerous other for-profit and nonprofit organizations to support the work of the School of Public Health and the university more broadly. This research was also supported by gifts from multiple anonymous philanthropic donors to the Indiana University Foundation. These competing interests do not alter our adherence to PLOS ONE policies on sharing data and materials.

## Results

For the antibody negative participants (n = 1029) and antibody positive participants (n = 47), we observed no significant differences in protective behavior engagement between those who were randomized to receive test results immediately or after 4 weeks. For the baseline antibody negative participants, we also observed no difference in seroconversion outcomes between conditions.

## Conclusions

We found that receiving antibody test results did not lead to significant behavior change in undergraduate students whether the SARS-CoV-2 antibody result was positive or negative.

## Introduction

College campuses feature environments that are high risk for SARS-CoV-2 transmission, characterized by indoor locations with close contact (e.g., dorms, classrooms) and academic and social mixing that results in high contact rates [1]. Although student populations are at lower risk for severe COVID-19 disease themselves, increased infections among younger populations are often associated with corresponding increases among older, higher-risk populations [2]. Understanding the motivations and drivers of practicing SARS-CoV-2 protective behaviors among university student populations has consequences for both campus infection prevention and control as well as the health of the broader community.

Many universities that pursued in-person education in 2020, and had the resources to do so, implemented SARS-CoV-2 preventive measures including mask mandates, isolation and quarantine procedures, reduced classroom density, online learning, regular testing regimes of active infections using RT-PCR and/or antigen tests, and vaccine mandates [3–5]. The main goals of regular testing were to control disease spread through quarantine of close contacts of individuals who tested positive and to track the prevalence of infection in the campus community. However, students may have understood a positive SARS-CoV-2 test to mean that they could relax preventive behaviors, assuming that a prior infection conferred immunologic protection. SARS-CoV-2 antibody tests, though less frequently used than tests of active infections, are another source of information about prior infection status that may influence risk perceptions.

Learning the results of a SARS-CoV-2 test may influence individual behavior through several plausible mechanisms. Theoretically, risk compensation postulates that individuals have some amount of risk that they are willing to assume and will change their behaviors to match that level of risk [6]. Relatedly, behavioral disinhibition theory posits that feelings of protection against one health concern may cause people to engage in behaviors that put them at risk for other health issues. A necessary condition for these behavioral pathways to operate with SARS-CoV-2 tests is that people must have working perceptions that a relationship between prior infection and protection against future infections exists. These perceptions are likely related to overall COVID-19 and immunological knowledge [7–9].

A number of public health interventions have been questioned for potentially causing risk compensation including wearing a bicycle helmet to prevent head injury [10, 11], seat belt mandates to prevent traffic fatalities [12], pre-exposure prophylaxis [13] and male circumcision [14] to prevent HIV infection, and HPV vaccination to prevent cervical cancer [15, 16].

However, rigorous randomized studies have shown that though an individual may make riskier choices, the interventions in question still have an overall beneficial effect on the population outcomes they are designed to prevent. Given the many influences on SARS-CoV-2 protective behaviors that may also correlate with seeking information about prior infection status, a randomized trial is the most appropriate design to avoid confounding by these factors. Understanding the impact of information about past SARS-CoV-2 infection on preventive behavior is essential to managing viral control and for learning more about expected behavior post natural infection and vaccination. The SARS-CoV-2 vaccine has an overwhelmingly beneficial effect on lowering risk of serious COVID-19 disease and mortality [17]. However, understanding how much, if any, risk compensation might occur after natural infection or vaccination has important potential consequences for disease control.

In this study, we aimed to answer the research question, 'Does learning the results of an antibody test change SARS-CoV-2 protective behaviors?', with a randomized control trial (RCT) in a sample of undergraduate students during the Fall 2020 semester. Aligned with our hypothesis that behavior change would be differential by antibody status (i.e., those with a positive antibody test may change their behavior in a way that would be different than those with a negative antibody test), we assessed results separately for antibody-positive and antibody-negative participants.

## Methods

### Study setting

This study was conducted among undergraduate students enrolled at Indiana University's (IU) flagship Bloomington campus in Fall 2020. IU Bloomington is a public university located in southern Indiana with a large undergraduate student population of around 33,000 students. During the Fall 2020 semester, IU Bloomington had strict COVID-19 protection measures in place, including a mask mandate, classroom and dorm de-densification to support physical distancing, restrictions on event sizes, and mandatory, random asymptomatic RT-PCR SARS-CoV-2 testing.

### Study participants and eligibility criteria

We conducted a simple random sample of all IU Bloomington undergraduate students enrolled at the beginning of the Fall 2020 semester yielding 7,499 students. Students in the sample were eligible to participate if they were (1) aged 18 years or older, (2) a current IU Bloomington undergraduate student, and (3) currently residing in Monroe County, Indiana. Of those sampled, 4,069 potential participants met the inclusion criteria for the study. All sampled students were contacted by email with a study invitation and a link to detailed information about study objective and procedures. The study team offered potential participants the opportunity for email or telephone consultations to answer any additional questions about study participation. After reviewing study information, interested and eligible students provided written informed consent remotely [18, 19] The IU Human Subjects and Institutional Review board provided ethical approval for this study protocol (Protocol #2008293852). We also prospectively registered this study protocol in the ClinicalTrials.gov database (#NCT04620798) and the data collection protocol was followed without any changes [20].

### Data collection

After enrolling in the study, all participants completed an online baseline survey capturing socio-demographic and behavioral data as well as information on SARS-CoV-2 testing and

infection history. Participants attended an in-person clinic visit for baseline antibody testing between September 14 and 30, 2020, and again for a second antibody test between November 8–14, 2020. Indiana University had a shortened semester for on-campus activities in Fall 2020, and this second set of dates aligned with the last weeks students were physically on-campus. Participants were instructed not to attend their in-person visits if they were experiencing COVID-19 symptoms, had tested positive for SARS-CoV-2 in the two weeks before their appointment, or had been directed to isolate or quarantine.

All antibody testing was conducted using the BGI Colloidal Gold IgM/IgG rapid assay kit [21]. Trained nursing staff, who were blinded to the randomized group, collected fingerstick blood samples, and the laminar flow test kits displayed results of antibody positivity for both IgG and IgM SARS-CoV-2 antibodies within 5 minutes. At the time of the visit, research staff read the antibody test results and immediately entered them into the study database. Randomized conditions and results delivered were determined by this initial test read. Participants did not learn their antibody test results at time of the baseline study visit. They were informed their results would be communicated to them by email within 4 weeks. At the randomly assigned time for participants to receive their results, they were emailed a secure link with password protection to access their results (positive or negative). This link also included a brief educational message about the importance of maintaining COVID-19 protective behaviors regardless of antibody test status (see S1 Fig).

We took several steps to improve and better understand the accuracy of the antibody test results. First, a team of trained research staff independently conducted a second review of the test results using high-resolution digital photographs of the test kits. Discrepancies between these assessments were resolved by consensus based on careful review of the digital photographs. Second, to assess the sensitivity and specificity of the antibody test kits, we conducted an independent laboratory assessment with banked hospital-based samples that were known to be SARS-CoV-2 antibody negative (n = 100) and known SARS-CoV-2 antibody positive blood samples (n = 94). The test kits correctly ascertained 100% (100/100) of the known negative samples, and 63.8% (60/94) of the known positive samples. This sensitivity is consistent with estimates from other laminar flow point of care antibody tests [22]. Importantly, the moderately low sensitivity of our antibody tests should not influence the results of this study given the randomized design and the objective to understand behavior change after receiving test results.

After the baseline antibody test, participants were followed-up in parallel every two weeks with online surveys assessing their engagement in key COVID-19 protective behaviors using a scale from the World Health Organization COVID-19 survey tool [23]. A total of four surveys were administered over the 8 weeks of follow-up. All online surveys were developed in and delivered using REDCap electronic data capture software [24, 25]. REDCap also supported all other data entry and database management aspects of the study. Participants were compensated up to $30 for completion of all study procedures.

## Randomization

Participants were randomly assigned to receive their baseline antibody test results either immediately (within 24 hours) or delayed (in 4 weeks). We used stratified block randomization, with a block size of 10, to obtain an equal number of participants in the two conditions (1:1 allocation) between strata of baseline SARS-CoV-2 antibody status. An independent statistician generated a random sequence using SAS software [26]. We programmed REDCap to randomly allocate participants to either the immediate or the delayed condition based on the allocation sequence at the time study staff entered baseline antibody test results. Allocation

was concealed from all investigators and field staff as it was not possible to predict or decipher the next allocation performed by REDCap.

## Key measures

Our <u>exposure</u> of interest was the randomized timing of antibody test result distribution: immediate (within 24 hours) versus delayed (4 weeks).

The <u>primary outcomes</u> of interest were engagement in four key COVID-19 protective behaviors two weeks after the baseline antibody test. In an on-line survey, participants were asked to quantify their level of engagement with the following behaviors in the past 7 days [23]: 1) avoiding social events, 2) staying at home from work/school, 3) wearing a face mask in public, and 4) ensuring physical distance in public. Response options were always, very often, sometimes, rarely, and never. All outcomes were dichotomized for primary analysis into: Always and Very Often versus Sometimes, Rarely, and Never. We conducted a sensitivity analysis to assess the robustness of our results to the choice of dichotomizing the behavioral outcomes. In this sensitivity analysis, we converted the Likert responses into a continuous variable by assigning numeric values (1 = never, 5 = always) to the responses and summing across all four protective behaviors (maximum possible value = 25).

As a <u>secondary outcome</u>, we measured SARS-CoV-2 seroconversion for participants who tested negative for SARS-CoV-2 antibodies at the baseline round of testing. If a participant tested newly positive for SARS-CoV-2 antibodies at the endline study visit after 8 weeks of follow-up, they were considered to have experienced the seroconversion outcome.

We also collected data on key covariates to characterize the study sample, and to stratify the sample for analysis. The variables we used to characterize the study sample were: age (in years), sex at birth (male/female), gender identity (man/woman/gender non-conforming), race (white/Asian/Black/multi-racial/other), school year, residence (on- or off-campus), and Greek organization affiliation (yes/no).

We used baseline antibody test result (positive/negative) to stratify the sample for analysis as we expected behavior change to be differential by antibody status. Participants with positive test results for either IgG or IgM antibodies were considered antibody-positive, while those negative for both IgG and IgM antibodies were considered antibody-negative. We expected participants who tested positive for SARS-CoV-2 antibodies to reduce protective behaviors after receiving results based on the belief that antibodies might provide protection against future infection. We expected participants who tested negative for SARS-CoV-2 antibodies to exhibit no change or a small increase in protective behaviors after receiving results given that a negative status could serve as a reminder that they are still susceptible to infection.

## Statistical analysis

We used Pearson's chi-square test to test the independence of demographic distributions between the two treatment conditions. We ensured that outcome categories were mutually exclusive and that all cells had values of at least 5.

We specified modified Poisson regression models to conduct intention-to-treat analyses, estimating risk ratios for the associations between randomly assigned treatment conditions (immediate vs. delayed antibody test results) and participation in four protective behaviors at two weeks from baseline. We specified separate models for each of the four behavioral outcomes: social event avoidance, staying home from work or school, mask use in public, and physical distancing in public. For the sensitivity analyses around the dichotomized outcome coding decision, we specified linear regression models to estimate the associations between randomly assigned treatment condition and the total of the sums of the Likert responses across

all four protective behaviors separately for those who tested positive and those who tested negative at baseline.

A modified Poisson regression model was also specified to compare the risk of SARS-CoV-2 seroconversion at the end of the trial (8 weeks after baseline) between immediate and delayed conditions, among the subsample of those with a negative antibody test result at baseline.

Analysts were blinded from the group allocation by enlisting an independent statistician to hold the key to which group (1 or 2) was randomized to immediate versus delayed test results. The reliability of the analysis was confirmed by a second independent analysis. Data were analyzed using SAS 9.4 statistical software [26]. All figures were plotted using R version 4.1.1 [27].

## Results

Between September 14, 2020 and September 30, 2020, 1,397 participants (response rate of 34.4%) consented to participate in this study from a sample of 4,069 randomly selected, eligible IU-B undergraduate students. We report here the results from 1,076 (77% of those who consented) who completed the baseline survey and baseline antibody test (Fig 1 and Table 1). The median age of participants was 20 years (IQR 19–21) and the ages of study participants largely aligned with traditional undergraduate student ages of 18–21 (90.6%). The majority of study participants identified as women (64%). The study sample was also majority white (79%), with lower representation of Asian (8%), Black (1%), and multi-racial (8%) students. Participants were fairly evenly distributed across the four years of traditional class standing (freshmen through seniors). About one-third (32%) of participants lived on-campus and about one-quarter (24%) were affiliated with Greek student organizations. As expected, none of these demographic variables differed significantly by randomized delayed versus immediate treatment condition.

Overall, 49 participants tested positive for SARS-CoV-2 antibodies at baseline (4.6%) and we observed 42 new seroconversions over the course of the 8 weeks of follow-up. The distribution of the four behavioral outcomes was generally balanced at baseline, with higher engagement reported for certain behaviors. Wearing a face mask (98.3% always or very often) and ensuring physical distancing (89.6% always or very often)] were more prevalent than staying home from work/school (43.2%) and avoiding social events (57.0%). The distribution of engagement in each behavior at baseline, 2-weeks (time of primary behavioral endpoints), 4-weeks, 6-weeks, and 8-weeks (endline) are displayed in Fig 2, stratified by randomized immediate vs. delayed treatment condition and by serostatus.

In the overall study sample, we found no significant differences between treatment conditions in any of the four behavioral outcomes (Table 1). Two weeks after antibody test results were reported to participants in the immediate results condition, chi-square tests indicated that participants in this condition did not report significantly higher or lower engagement in wearing face masks, staying home from work and school, avoiding social events, or ensuring physical distancing in public. Similarly, no significant differences were observed between study conditions for the seroconversion outcome, with 22 delayed condition and 20 immediate condition participants experiencing seroconversion (6% vs. 5% respectively).

Regression models stratified by baseline serostatus did not reveal any significant behavioral differences by study condition (Table 2) for participants receiving either positive or negative antibody test results. Taking face mask use as a representative example, for seronegative participants, receiving antibody test results was not associated with higher or lower face mask engagement [RR (95% CI): 1.01 (1.00, 1.03)]. Similar results were observed for our smaller sample of seropositive participants [RR (95% CI): 0.91 (0.80, 1.04)]. We also did not observe significant differences in seroconversion risk by timing of antibody test results among those

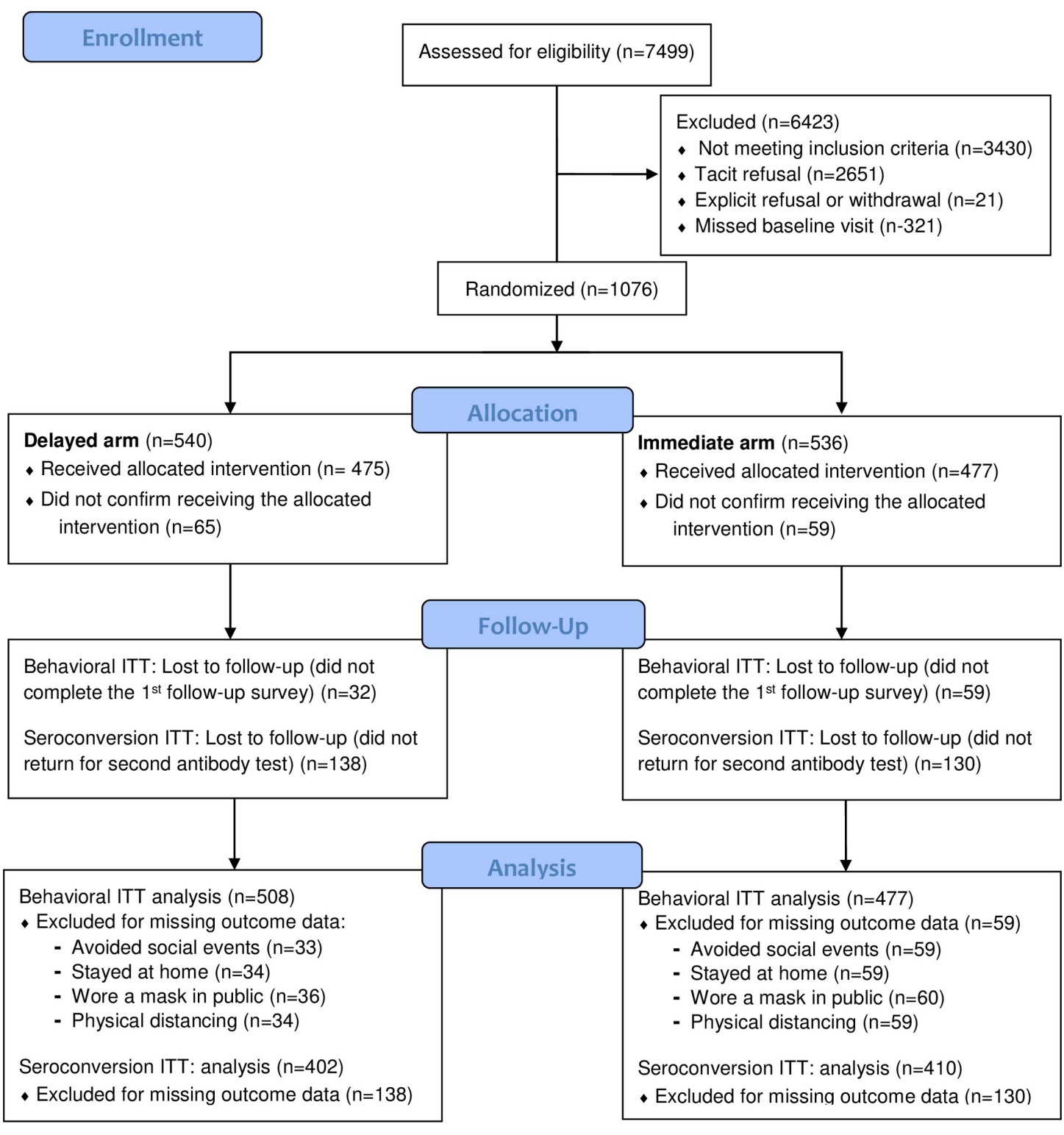

**Fig 1. Consort participant flow diagram.**

who were seronegative at baseline. Participants in the immediate condition did not exhibit a higher or lower risk of seroconversion at 8 weeks, compared to the delayed condition [RR

**Table 1. Characteristics of the n = 1076 Indiana University undergraduate students enrolled in the trial, Fall 2020.**

|  | Total (n = 1076) | Delayed condition (n = 540) | Immediate condition (n = 536) | P-value |
|---|---|---|---|---|
|  | N (%) | N (%) | N (%) |  |
| *Sociodemographic covariates* | | | | |
| **Age** |  |  |  | 0.766 |
| 18 years old | 208 (20.6) | 97 (19.3) | 111 (21.9) |  |
| 19 years old | 224 (22.2) | 115 (22.9) | 109 (21.5) |  |
| 20 years old | 228 (22.6) | 119 (23.7) | 109 (21.5) |  |
| 21 years old | 255 (25.3) | 123 (24.5) | 132 (26.0) |  |
| 22+ years old | 95 (9.4) | 48 (9.6) | 47 (9.3) |  |
| Missing | n = 66 | n = 38 | n = 28 |  |
| **Sex at birth** |  |  |  | 0.179 |
| Male | 382 (35.7) | 181 (33.7) | 201 (37.6) |  |
| Female | 689 (64.3) | 356 (66.3) | 333 (62.4) |  |
| Missing | n = 5 | n = 3 | n = 2 |  |
| **Gender identity** |  |  |  | 0.162 |
| Male | 379 (35.4) | 179 (33.3) | 200 (37.5) |  |
| Female | 680 (63.5) | 354 (65.9) | 326 (61.1) |  |
| Non-conforming | 12 (1.1) | 4 (0.7) | 8 (1.5) |  |
| Missing | n = 5 | n = 3 | n = 2 |  |
| **Race** |  |  |  | 0.938 |
| Asian | 80 (7.5) | 37 (6.9) | 43 (8.1) |  |
| Black | 13 (1.2) | 7 (1.3) | 6 (1.1) |  |
| Multi-racial | 85 (7.9) | 42 (7.8) | 43 (8.1) |  |
| Other | 46 (4.3) | 22 (4.1) | 24 (4.5) |  |
| White | 847 (79.1) | 430 (79.9) | 417 (78.2) |  |
| Missing | n = 5 | n = 2 | n = 3 |  |
| **Undergraduate school year** |  |  |  | 0.514 |
| First year | 236 (22.1) | 113 (21) | 123 (23.1) |  |
| Second year | 246 (23) | 127 (23.7) | 119 (22.3) |  |
| Third year | 264 (24.7) | 137 (25.5) | 127 (23.8) |  |
| Fourth year | 297 (27.8) | 143 (26.6) | 154 (28.9) |  |
| Fifth year or more | 27 (2.5) | 17 (3.2) | 10 (1.9) |  |
| Missing | n = 6 | n = 3 | n = 3 |  |
| **Residence** |  |  |  | 0.326 |
| Off-campus | 733 (68.4) | 375 (69.8) | 358 (67.0) |  |
| On-campus | 338 (31.6) | 162 (30.2) | 176 (33.0) |  |
| Missing | n = 5 | n = 3 | n = 2 |  |
| **Greek affiliation status** |  |  |  | 0.918 |
| No | 812 (75.9) | 409 (76.0) | 403 (75.8) |  |
| Yes | 258 (24.1) | 129 (24.0) | 129 (24.3) |  |
| Missing | n = 6 | n = 2 | n = 4 |  |
| *Outcomes* | | | | |
| **Avoided social events (2 weeks)** |  |  |  | 0.437 |
| Always or Very Often | 445 (49.6) | 236 (50.9) | 209 (48.3) |  |
| Never, Sometimes, or Rarely | 452 (50.4) | 228 (49.1) | 224 (51.7) |  |
| Not Applicable | n = 87 | n = 43 | n = 44 |  |
| Missing | n = 92 | n = 33 | n = 59 |  |
| **Stayed at home from work/school (2 weeks)** |  |  |  | 0.172 |

*(Continued)*

**Table 1.** (Continued)

| | Total (n = 1076) | Delayed condition (n = 540) | Immediate condition (n = 536) | P-value |
|---|---|---|---|---|
| | N (%) | N (%) | N (%) | |
| Always or Very Often | 337 (39.8) | 185 (42.1) | 152 (37.4) | |
| Never, Sometimes, or Rarely | 509 (60.2) | 255 (58.0) | 254 (62.6) | |
| Not Applicable | n = 137 | n = 66 | n = 71 | |
| Missing | n = 93 | n = 34 | n = 59 | |
| **Wore a mask in public (2 weeks)** | | | | 0.269 |
| Always or Very Often | 963 (98.3) | 493 (97.8) | 470 (98.7) | |
| Never, Sometimes, or Rarely | 17 (1.7) | 11 (2.2) | 6 (1.3) | |
| Not Applicable | n = 0 | n = 0 | n = 0 | |
| Missing | n = 96 | n = 36 | n = 60 | |
| **Physical distancing in public (2 weeks)** | | | | 0.923 |
| Always or Very Often | 879 (89.4) | 452 (89.3) | 427 (89.5) | |
| Never, Sometimes, or Rarely | 104 (10.6) | 54 (10.7) | 50 (10.5) | |
| Not Applicable | n = 0 | n = 0 | n = 0 | |
| Missing | n = 93 | n = 34 | n = 59 | |
| **Seroconversion (8 weeks)** | | | | 0.726 |
| No | 766 (94.8) | 380 (94.5) | 386 (95.1) | |
| Yes | 42 (5.2) | 22 (5.5) | 20 (4.9) | |
| Not Applicable | n = 49 | n = 24 | n = 25 | |
| Missing | n = 219 | n = 114 | n = 105 | |

(95% CI): 0.94 (0.52, 1.71)]. Assumptions for all modified Poisson models were assessed and met. In the sensitivity analysis using the summed Likert responses as the outcome, we also did not find any significant differences between protective behaviors at 2-week follow-up between those who received test results immediately compared to those randomized to the delayed arm (S1 Table and S2 Fig).

## Discussion

In this study, we found no evidence for risk compensation among undergraduate students after receiving SARS-CoV-2 antibody test results. Students who were randomized to receive their antibody test results immediately did not report engaging in different levels of COVID-19 risk behaviors compared to students who were randomized to receive their antibody test results later. This lack of association held across all four behaviors we examined (staying home from school/work, avoiding social events, wearing face masks, and physical distancing), and for the SARS-CoV-2 seroconversion outcome. Importantly, stratified analysis by baseline antibody serostatus did not reveal differences in findings by whether participants received a positive or negative antibody test result.

That we found that behavior remained largely unchanged after receipt of antibody test results aligns with findings from prior studies of risk compensation in other settings. Five interventions are often recognized for their potential to cause risk compensation: bicycle helmets, seatbelts, voluntary medical male circumcision and pre-exposure prophylaxis for HIV prevention, and HPV vaccination. Exhaustive reviews in each of these areas have consistently concluded that there is, in fact, little evidence for increased risk-taking after intervention exposure [10–13, 16, 28]. In spite of the weak record for risk compensation in other settings, many have raised concerns about the threat of risk compensation in the COVID-19 pandemic [29]. Vaccines and face mask mandates have been two areas of concern for risk compensation, but

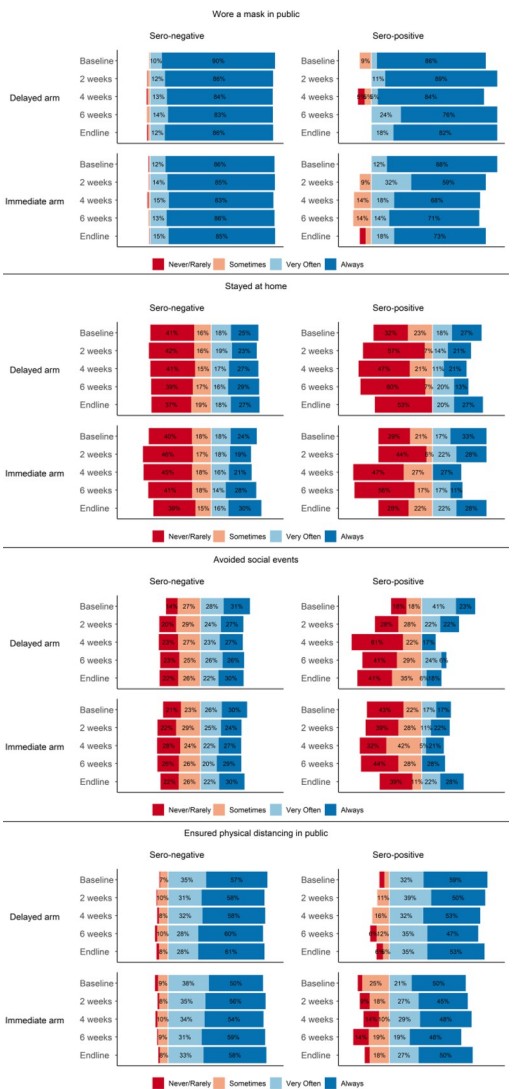

**Fig 2. Protective behaviors by baseline sero-status in delayed and immediate randomized arms.**

very few empirical studies have tested these associations. However, there is at least some suggestive evidence that risk compensation may, in fact, occur in settings with face mask mandates [30]. With the lack of risk compensation observed in our study, we provide additional empirical data to better shape our understanding of behavioral disinhibition in the COVID-19 pandemic.

The context of the study setting and study design have important implications for interpretation of our findings. First, changes in risk perception among those who tested positive for antibodies may not have translated into changed behaviors due to COVID preventive measures taken at the university and community level. Indiana University and Monroe County, where it is located, both had mask mandates and bans on large group social gatherings during the study period. Second, in some social circles, social norms reinforced protective behaviors as being the socially responsible course of action. Those external restrictions on behavior may exert a larger influence than an internal assessment of risk. Third, study participants were provided clear guidance to continue to practice COVID-19 protective behaviors regardless of

**Table 2. Associations between treatment condition (immediate vs delayed antibody test results) and key behavioral and biological outcomes, stratified by baseline antibody status.**

| | 2 weeks | 8 weeks | 2 weeks | 8 weeks |
|---|---|---|---|---|
| Outcomes | RR (95% CI)[1] | RR (95% CI)[1] | RD (95% CI)[1] | RD (95% CI)[1] |
| *Negative antibody test results (n = 1029)*[2] | | | | |
| Social event avoidance[3] | 0.96 (0.84, 1.09) | | -0.02 (-0.09, 0.04) | |
| Staying home from work/school[3] | 0.87 (0.74, 1.04) | | -0.05 (-0.12, 0.01) | |
| Mask use in public[3] | 1.01 (1.00, 1.03) | | 0.01 (-0.002, 0.03) | |
| Physical distancing in public[3] | 1.01 (0.97, 1.06) | | 0.01 (-0.03, 0.05) | |
| Seroconversion | - | 0.94 (0.52, 1.71) | - | -0.003 (-0.03,0.03) |
| *Positive antibody test results (n = 47)*[2] | | | | |
| Social event avoidance[3] | 0.75 (0.33, 1.72) | | -0.11 (-0.43, 0.21) | |
| Staying home from work/school[3] | 1.40 (0.60, 3.25) | | 0.14 (-0.20, 0.48) | |
| Mask use in public[3] | 0.91 (0.80, 1.04) | | -0.09 (-0.21, 0.03) | |
| Physical distancing in public[3] | 0.82 (0.60, 1.11) | | -0.16 (-0.40, 0.07) | |
| Seroconversion | - | - | - | - |

[1]Reference group refers to delayed antibody test results condition

[2]Wald p-values for the interaction between serostatus and intervention arm were: 0.57 (social event avoidance), 0.28 (staying home from work/school), 0.11 (mask use in public), and 0.18 (physical distancing in public)

[3]Comparing the probability of "Very Often/Always" vs. all other response categories

their antibody test results (see S1 Fig), which may also account for our findings of no observable behavior change. Relatedly, it is possible that risk behaviors did not differ between study arms because participants did not otherwise have strong perceptions of a relationship between serostatus and protection against future infections. Finally, as results were disseminated by email, we cannot confirm that everyone in each treatment condition received their results on the expected schedule, or at all. However, operational and survey data we collected indicated that participants received their test results at similar rates in both arms, alleviating concerns of differential intervention uptake (see Fig 1).

Results from this study likely do not generalize to university settings with different COVID-19 policies in place. However, the results are likely to be generalizable to other young adult populations at other large, public universities similar to the one where this study was conducted. We used random sampling from the IU student population and enrolled a large sample into the randomized trial. Though we observed a relatively low response rate, it is comparable to other response rates in university settings [31]. Nevertheless, to assume that our study population stands in for our larger target population, we have to assume that the enrolled study population had similar risk compensation responses as those who did not enroll. Undergraduate women, in particular, were overrepresented in our study population relative to the student body. Other demographic variables, however, generally tracked with those observed in the student body. After enrollment, we maintained a relatively high follow-up rate for the 2-week behavioral outcomes (>90%), but had more limited completeness of follow-up for the secondary seroconversion outcome at 8-weeks (79%). Reasons for the higher attrition at this later timepoint may have involved the higher burden and stricter COVID protocols with the in-person study visit required for the serology testing, and the timing at the end of the in-person semester coinciding with more limited availability of our student population.

Inference from this study is strengthened by the rigorous randomized study design used. By randomly assigning half of our participants to receive their results immediately, we were able to isolate the effect of receiving the results and minimize the potential for confounding to bias our

results. Using a randomized trial design is critical because seeking testing for SARS-CoV-2, either for acute infection [32] or antibody testing, requires both personal motivation and access to services which render comparisons between those who receive testing and those who do not subject to significant possible bias. These factors, and those related to them, would be major sources of confounding in an observational study comparing those who did and did not receive antibody tests. However, although we had a large sample overall (n = 1076), only 47 participants tested positive for SARS-CoV-2 antibodies at baseline, limiting the precision with which we measured the trial effects in that subgroup. The relatively low number of positive tests we observed could be accurately capturing the low seropositivity at this early stage of the pandemic (September-November 2020). Negative antibody tests would also be expected for participants who have experienced a SARS-CoV-2 infection but not yet developed antibodies, or whose antibodies have already waned below detectable levels. Of course, some of our study participants may have falsely tested negative, due to the moderately low test sensitivity (64%) or researcher error in administering or reading the tests. In both of these scenarios, the measurement error is unlikely to be differential by trial arm as the study team was blinded to trial arm allocation.

Our findings provide reassuring evidence that the receipt of SARS-CoV-2 antibody test results are unlikely to cause major changes in protective behaviors, with the assumption that the results generalize beyond a university student population. Although individual risk perceptions may be updated with information on antibody positivity or antibody negativity status, those updated risk perceptions do not appear to result in risk compensation. These findings may also be informative to other experiences or interventions in the context of the COVID-19 pandemic such as natural infection, vaccination, and mask mandates. Future work on COVID-19 risk compensation should focus on extending our findings with these COVID-relevant exposures, in different populations, and by covering updated time periods in the pandemic.

## Supporting information

**S1 Fig. Educational material provided to participants at time of antibody test result return.**
(TIF)

**S2 Fig. Sensitivity analysis for association between treatment condition (immediate vs delayed antibody test results) and mean frequency of engagement in protective behaviors at 2 weeks (primary endpoints), and across additional timepoints over 8 weeks of follow-up, stratified by baseline antibody status.**
(TIF)

**S1 Table. Sensitivity analysis for association between treatment condition (immediate vs delayed antibody test results) and mean frequency of engagement in protective behaviors at 2 weeks, stratified by baseline antibody status.**
(DOCX)

**S1 File. Deidentified minimal dataset.**
(CSV)

**S2 File. SAS code to produce study results.**
(SAS)

**S1 Data.**
(DOCX)

**S1 Checklist.**
(PDF)

## Author Contributions

**Conceptualization:** Christina Ludema, Molly S. Rosenberg, Jonathan T. Macy, David B. Allison.

**Data curation:** Sina Kianersi, Maya Luetke, Chen Chen.

**Formal analysis:** Lilian Golzarri-Arroyo, Erin Ables.

**Funding acquisition:** Kevin Maki, David B. Allison.

**Methodology:** Christina Ludema, Molly S. Rosenberg.

**Project administration:** Christina Ludema, Molly S. Rosenberg, Jonathan T. Macy, Sina Kianersi, Maya Luetke.

**Supervision:** Christina Ludema, Molly S. Rosenberg.

**Validation:** Erin Ables.

**Visualization:** Lilian Golzarri-Arroyo.

**Writing – original draft:** Christina Ludema, Molly S. Rosenberg.

**Writing – review & editing:** Christina Ludema, Molly S. Rosenberg, Jonathan T. Macy, Sina Kianersi, Maya Luetke, Chen Chen, Lilian Golzarri-Arroyo, Erin Ables, Kevin Maki, David B. Allison.

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
