## [Decision Letter · Decision Letter 0]

4 Sep 2022

PONE-D-22-18389No evidence for risk compensation in undergraduate students after SARS-CoV-2 antibody test results: a randomized controlled trialPLOS ONE

Dear Dr. Rosenberg,

Thank you for submitting your manuscript to PLOS ONE. After careful consideration, we feel that it has merit but does not fully meet PLOS ONE’s publication criteria as it currently stands. Therefore, we invite you to submit a revised version of the manuscript that addresses the points raised during the review process.

please submit your revised manuscript by Oct 19 2022 11:59PM. If you will need more time than this to complete your revisions, please reply to this message or contact the journal office at plosone@plos.org. Please include the following items when submitting your revised manuscript:A rebuttal letter that responds to each point raised by the academic editor and reviewer(s). You should upload this letter as a separate file labeled 'Response to Reviewers'.A marked-up copy of your manuscript that highlights changes made to the original version. You should upload this as a separate file labeled 'Revised Manuscript with Track Changes'.An unmarked version of your revised paper without tracked changes. You should upload this as a separate file labeled 'Manuscript'.

We look forward to receiving your revised manuscript.

Kind regards,

Hermano Alexandre Lima Rocha

Academic Editor

PLOS ONE

Journal Requirements:

   "CL, MR, JT, SK, ML, CC, LGA, and EA have declared that no competing interests exist.

We have read the journal's policy and KM has the following competing interests:

1. Consulting regarding development of therapeutics for Covid-19 (not related to the current manuscript) and payments made to a private clinic in which I am a partner for conduct of a Covid treatment trials (not related to the current manuscript) 

2. Payments made to a private clinic in which I am a partner for conduct of a vaccine trial (not related to the current manuscript) "

Additional Editor Comments:

Dear authors

Hope you are doing well

Please follow reviewers suggestions and send the manuscript back to us with the required amendments and a letter letting us know the your responses.

Best wishes

Reviewers' comments:

Reviewer's Responses to Questions

**Comments to the Author**

1. Is the manuscript technically sound, and do the data support the conclusions?

Reviewer #1: Yes

Reviewer #2: Yes

Reviewer #3: Partly

2. Has the statistical analysis been performed appropriately and rigorously? 

Reviewer #1: Yes

Reviewer #2: Yes

Reviewer #3: Yes

3. Have the authors made all data underlying the findings in their manuscript fully available?

Reviewer #1: No

Reviewer #2: No

Reviewer #3: Yes

4. Is the manuscript presented in an intelligible fashion and written in standard English?

Reviewer #1: Yes

Reviewer #2: Yes

Reviewer #3: Yes

5. Review Comments to the Author

Reviewer #1: No main concerns on this article. Some comments:

- informed consent process: the remote consent administration may not be as effective for transferring correct information to students; probably in this case a telephone contact to better understand students' perception and knowledge of COVID would have been more effective

- apart from test accuracy, could you comment on the understanding of the relation between a positive test and the real protection from COVID infection? Students may not be aware of this relation and this relation is not so strong

- could you comment on the reasons of 23% of students lost to FUP?

- could you comment on the small number of participants tested positive?

Reviewer #2: In general, the manuscript is well-written and describes an interesting study of risk compensation. The following represent relatively minor criticisms.

It would be most useful to evaluate the data and code used for analysis.

* Analysis issues:

The analysis is performed on dichotomized Likert scale responses. The protocol does specify this analysis, but does not specify the dichotomization criteria prospectively. Given the distribution of the data, it does seem unlikely that the choice of dichotomization would matter, but a more principled approach to the analysis would respect the specific structure of the response. After all, if the data were not going to be analyzed as Likert scaled values, then why collect them as such? Alternatively, the protocol should have pre-specified the dichotomization criteria.

The authors do present an alternative analysis treating the Likert scale responses as interval scaled as a sensitivity analysis. This is comforting but does not necessarily completely ameliorate the issue.

A reasonable analysis for these data would be an ordinal logistic regression analysis. A basic approach would be to use a proportional odds model (McCullagh and Nelder, 1989). McCullagh and Nelder also present a slight extension useful for testing the proportional odds assumption versus monotonically changing odds.

For an excellent exposition on the nuts and bolts of implementing the proportional odds model, assessing goodness of fit, and interpreting the results, see Harrell (2010). Professor Harrell has also made numerous resources available online.

Furthermore, is there any specific reason not to include all data in a single analysis with two factors (seropositivity status and intervention) and their interaction effect? The imbalance might mitigate against this, but it would still be of interest to evaluate the interaction term.

And, along those lines, although it is probably only feasible (if at all) with the dichotomized version of the data (or with the Likert scale data treated as interval scaled), a full model would consider three fixed effects (seropositivity status, intervention, and time) and their interactions. However, this model would require a method for handling the repeated measures across time for each subject. This could be carried out using the 'nlme' R package, most likely (Pinheiro and Bates, 2017). However, convergence is always an issue for these binomial responses. But, treating the responses as intervals scaled should meet with no issues. There are several R packages that extend this modeling in a Bayesian framework as well as other frameworks such as STAN (Carpenter et al, 2017) which could also be used to perform such modeling.

An interesting alternative approach to the sensitivity analysis could be carried out using the concept of specification curve analysis (Simonsohn, Simmons, and Nelson, 2020). In a nutshell, the idea is to specify all of the alternative paths that the analysis could have taken to assess the impact of the one path that was actually chosen. It would be quite feasible to implement this analysis using the 'specr' R package (Masur and Scharkow, 2020).

* Interpretation issues

The study sample represents a subset of students who self-selected to join the study (34.4% of randomly selected, eligible IU-B undergraduate students). It could be that in this subset there is no observed risk compensation, but that in the complementary subset of students that did not self-select into the study there is some observed risk compensation. To be able to widen the scope of inference likely requires some assumptions about the comparability of these two subsets. There seems to be no real way to avoid this so it must be addressed in the limitations.

The authors state "However, the results are likely to be generalizable to other young adult populations at other predominantly white universities similar to the one where this study was conducted." (lines 339-340). Is race a deciding factor here? If so, please provide some citations that would support this.

Not only did states differ in their approaches to handling COVID-19 issues broadly, but even cities within states differed, as did universities within states. And, students at different universities could come from a more or less heterogeneous background. It seems as though all of these effects make it more difficult to generalize rather than more easy.

The authors mention "potential for confounding" in line 347. Please list these potential issues, if only briefly or at a high level.

In Figure 2, please show all available data rather than only three time points. Also, the bars in these figures appear offset in a way that makes them hard to interpret.

Although the stacked bar approach is not ideal, it is hard to suggest an alternative. One might attempt grouping bar charts but this is not likely to end well.

Given the linear model approach that was executed in the appendix, please also provide a plot of the least squares estimates over time for each combination of seropositivity status and intervention.

* References

Carpenter, B., Gelman, A., Hoffman, M.D., Lee, D., Goodrich, B., Betancourt, M., Brubaker, M., Guo, J., Li, P. and Riddell, A., 2017. Stan: A probabilistic programming language. Journal of statistical software, 76(1).

Harrell, F. E. (2010). Regression Modeling Strategies: With Applications to Linear Models, Logistic Regression, and Survival Analysis. Springer Series in Statistics.

Masur, P. K. and Scharkow, M. (2020). specr: conducting and visualizing specification curve analyses. https://cran.r-project.org/web/packages/specr/

McCullagh, P. and Nelder, J. A. (1989). Generalized linear models.

Pinheiro, J., Bates, D., DebRoy, S., Sarkar, D., Heisterkamp, S., Van Willigen, B., & Maintainer, R. (2017). Package ‘nlme’. Linear and nonlinear mixed effects models, version, 3(1).

Simonsohn, U., Simmons, J. P., and Nelson, L. D. (2020). Specification curve analysis. Nature Human Behaviour, 4(11), 1208-1214.

Reviewer #3: Thank you for asking me to review this article. Highly transmittable infectious diseases, such as COVID-19 are public health emergencies of international concern. There is still no definitive cure for some of those highly transmittable illness. Immunization and breaking the chain of infection is the only successful approach to mitigate its spread. However, while on one side immunization coverage is conditioned by the people’s acceptance of these vaccines, on the other side natural infection or vaccination has important potential consequences on preventive startegies and for disease control. In this context, aim of the paper under review is to assess whether objective information about antibody status, particularly for those who are antibody negative and likely still susceptible to SARS-CoV-2 infection, increases protective behaviors and, moreover, assessing whether a positive antibody test results in decreased protective behaviors.

The subject under study is certainly important, especially in the historical period we are experiencing. The article presents interesting results but it must be further improved.

Title: it can be improved, highlight the object of the study.

Abstract. I encourage the authors to add more detail about their core contributions in the abstract.

Introduction: The authors should make clearer what is the gap in the literature that is filled with this study. The authors must better frame their study within the vast body of literature that also addressed the issue of knowledge concerning COVID-19 that can affect the implementation of control measures in different groups of population (refer to articles with DOI: https://doi.org/10.3390/ijerph182010872).

Methods: The survey was conducted using a non-standard tool. The use of an unreliable instrument is a serious and irreversible limitation. A validation process must be performed to evaluate the tool. What about reliability, intelligibility and validation index? Was a pilot study performed?

The enrolment procedure must be specified. How did the authors choose the way to select the sample? This can represent a great bias origin. How did they avoid the selection bias? The authors do not propose a minimum sample size. Without the numerical identification of the reference population is not clear the validity of the study. A non-representative sample is by its self a non-sense-survey.

Statistical analysis: I suggest to insert a measure of the magnitude of the effect for the comparisons. Please consider to include effect sizes.

Discussion: I also suggest expanding. Emphasize the contribution of the study to the literature. The discussion must be updated with the discussion regarding knowledge about the diseases (see the above mentioned reference) that can be an important confounding factors for this study. The Authors should add more practical recommendations for the reader, based on their findings. Also, the section of limitations and future search is also very short, the Authors could elaborate on that.

6. PLOS authors have the option to publish the peer review history of their article (what does this mean?). If published, this will include your full peer review and any attached files.

Reviewer #1: **Yes: **Manuela Monti

Reviewer #2: No

Reviewer #3: No

---

## [Author Response · Author response to Decision Letter 0]

26 Oct 2022

We thank the reviewers for their valuable and constructive suggestions how to improve our paper “Does receiving a SARS-CoV-2 antibody test result change COVID-19 protective behaviors? Testing risk compensation in undergraduate students with a randomized controlled trial” (Ms. No.: PONE-D-22-18389). Below, we explain point-by-point how we have addressed each of the reviewers’ comments in the revised version of our paper. 

Editorial Comments

Response: We have now aligned our manuscript and references with the PLOS ONE style template. Thank you!

 "CL, MR, JT, SK, ML, CC, LGA, and EA have declared that no competing interests exist.

We have read the journal's policy and KM has the following competing interests:

1. Consulting regarding development of therapeutics for Covid-19 (not related to the current manuscript) and payments made to a private clinic in which I am a partner for conduct of a Covid treatment trials (not related to the current manuscript) 

2. Payments made to a private clinic in which I am a partner for conduct of a vaccine trial (not related to the current manuscript) "

 Thank you for the guidance around competing interests.

Response: We have updated our competing interests statement as requested in the attached cover letter. Please note that we have also added a competing interests statement for coauthor DBA. 

We confirm here and in the cover letter that the competing interests listed for KM and DBA do not alter our adherence to PLOS ONE policies.

Response: We have now created and included a deidentified minimal dataset with this submission in file ‘S1 File.’

We note this change in the manuscript and the cover letter and send our thanks for updating the data availability statement on our behalf.

Response: We have now made these edits to align our supporting information files with the journal naming requirements. 

R1 Comments

1. informed consent process: the remote consent administration may not be as effective for transferring correct information to students; probably in this case a telephone contact to better understand students' perception and knowledge of COVID would have been more effective

Response: We agree with the reviewer about the importance of excellent communication in the informed consent process. We used a remote consent procedure that was reviewed and approved by our Institutional Review Board. We want to assure the reviewer that we took several additional steps to communicate about the study with potential participants. Our study team offered telephone or email consultations with potential study participants on demand, to ensure questions about study participation were answered and that participants understood the relative risks and benefits of study participation. Our protocol also included active solicitation of participant questions at all in-person study visits (prior to serology testing).

Although remote consenting protocols do come with some challenges, a growing body of literature indicates that remote consenting can facilitate strong participant understanding (see, for example, citations below).

1.Rothwell, Erin, et al. "A randomized controlled trial of an electronic informed consent process." Journal of Empirical Research on Human Research Ethics 9.5 (2014): 1-7.

2.De Sutter, Evelien, et al. "Implementation of electronic informed consent in biomedical research and stakeholders’ perspectives: systematic review." Journal of medical Internet research 22.10 (2020): e19129.

We now cite this supporting literature in our ‘Study participants and eligibility criteria’ section, and have added information about the additional communication modes to which potential participants had access: 

“All sampled students were contacted by email with a study invitation and a link to detailed information about study objective and procedures. The study team offered potential participants the opportunity for email or telephone consultations to answer any additional questions about study participation. After reviewing study information, interested and eligible students provided written informed consent remotely with an electronic signature.[15, 16] The IU Human Subjects and Institutional Review board provided ethical approval for this study protocol (Protocol #2008293852).”

2. apart from test accuracy, could you comment on the understanding of the relation between a positive test and the real protection from COVID infection? Students may not be aware of this relation and this relation is not so strong 

Response: This is a great question. Our study, to our knowledge, is the first to assess the potential for risk compensation after SARS-CoV-2 antibody testing, and we did not specifically query perceptions around the link between serostatus and risk of infection in our survey. So, a possible explanation for the null effects we observed is that participants did not actually have strong perceptions of the relationship between serostatus and risk for future infection. This would be one possible reason why risk compensation was not observed. We have now added a description of this potential explanation to the third paragraph of the discussion:

“…, study participants were provided clear guidance to continue to practice COVID-19 protective behaviors regardless of their antibody test results (see S1 Fig), which may also account for our findings of no observable behavior change. Relatedly, it is possible that risk behaviors did not differ between study arms because participants did not otherwise have strong perceptions of a relationship between serostatus and protection against future infections…”

3. could you comment on the reasons of 23% of students lost to FUP? 

Response: Thanks for the opportunity to clarify. Our primary behavioral outcomes were assessed via electronic survey 2 weeks after baseline. At this time point, we maintained over 91% of our sample (missing n=92, 8.5%). We were not able to collect reasons for missing this survey wave, but we hope that this relatively low missing rate reassures the reviewer about the sensitivity of our results to missing data.

It was only for our secondary biologic outcome that we used the endline antibody test data to calculate seroconversion outcomes for participants after 8 weeks. For this outcome, the missingness rate was higher at 21%. Note that this excludes participants who tested positive for SARS-CoV-2 antibodies at baseline as they were not at risk for seroconversion (n=49). Although we did not collect data on reasons for missing the second antibody testing visit, we can speculate on a couple of contextual explanations: 1) This required an in-person visit for fingerprick, a higher burden than an electronic survey, 2) We requested that participants with COVID symptoms or under COVID quarantine/isolation to reschedule, but only had limited openings for these appointments, 3) The window for the 8 week study visit was in the last 2 weeks of the modified 2020 Fall semester with in-person instruction at Indiana University. Participants may have had competing activities that limited their availability for in-person study visit (exams, parties, preparation for moving back home, etc).

To address this important point, we have added the following text to the third paragraph of the discussion:

“After enrollment, we maintained a relatively high follow-up rate for the 2-week behavioral outcomes (>90%), but had more limited completeness of follow-up for the secondary seroconversion outcome at 8-weeks (79%). Reasons for the higher attrition at this later timepoint may have involved the higher burden and stricter COVID protocols with the in-person study visit required for the serology testing, and the timing at the end of the in-person semester coinciding with more limited availability of our student population.”

4. could you comment on the small number of participants tested positive? 

Response: Thanks for this question. It was helpful for us to reflect on the possible explanations for the relatively low seropositivity at baseline (n=49) and number of new seroconversions under observation (n=42). 

These numbers are somewhat low, but it is important to place them in the context of the early stage of the pandemic (September-November 2020), a time when this student population was just returning back to campus for the first time since March 2020. It is possible that our numbers are accurately capturing the relatively low seropositivity. Other ‘true negative’ reasons that could contribute to our relatively low observed seropositivity are that some participants could have experienced a SARS-CoV-2 infection, but not yet developed antibodies, or whose antibodies had already wanted below detectable results. 

Of course, there is also the possibility that some of our participants received false negative test results. This could have happened due to the moderately low antibody test sensitivity (64%), or if there was researcher error in reading or administering the tests.

To better characterize these potential explanations in the manuscript, we have now added the following text to the 5th paragraph of the discussion:

“The relatively low number of positive tests we observed could be accurately capturing the low seropositivity at this early stage of the pandemic (September-November 2020). Negative antibody tests would also be expected for participants who have experienced a SARS-CoV-2 infection but not yet developed antibodies, or whose antibodies have already waned below detectable levels. Of course, some of our study participants may have falsely tested negative, due to the moderately low test sensitivity (64%) or researcher error in administering or reading the tests. In both of these scenarios, the measurement error is unlikely to be differential by trial arm as the study team was blinded to trial arm allocation.”

R2 Comments

1. It would be most useful to evaluate the data and code used for analysis. 

Response: Thanks for flagging this for us. With our revised submission, we have now included a deidentified minimal dataset (‘S1 file.csv’) and code (‘S2 File.sas’) underpinning our analyses.

2. The analysis is performed on dichotomized Likert scale responses. The protocol does specify this analysis, but does not specify the dichotomization criteria prospectively. Given the distribution of the data, it does seem unlikely that the choice of dichotomization would matter, but a more principled approach to the analysis would respect the specific structure of the response. After all, if the data were not going to be analyzed as Likert scaled values, then why collect them as such? Alternatively, the protocol should have pre-specified the dichotomization criteria.

The authors do present an alternative analysis treating the Likert scale responses as interval scaled as a sensitivity analysis. This is comforting but does not necessarily completely ameliorate the issue.

A reasonable analysis for these data would be an ordinal logistic regression analysis. A basic approach would be to use a proportional odds model (McCullagh and Nelder, 1989). McCullagh and Nelder also present a slight extension useful for testing the proportional odds assumption versus monotonically changing odds.

For an excellent exposition on the nuts and bolts of implementing the proportional odds model, assessing goodness of fit, and interpreting the results, see Harrell (2010). Professor Harrell has also made numerous resources available online.

Response: As suggested, we ran proportional odds regression models for our four behavioral outcomes by serostatus. The proportional odds assumption was met for all models with the exception of the mask use outcome in the seropositive stratum (n=47). The p-value associated with the Score Test for this outcome/stratum was <0.0001. We suspect this was due in part to the limited sample size overall and within the less frequent response options for masking.

For the remaining outcome models that met the proportional odds assumption (score test p-value >0.05), we ran the proportional odds models to produce summary odds ratios to compare the odds of responding across the Likert categories. The proportional odds models we were able to run supported our primary results and the OLS sensitivity analysis results. We did not observe any statistically significant differences in behavioral outcomes by intervention arm:

Negative antibody test results (n=1029) 

Social event avoidance: OR 0.98 (0.73, 1.32)

Staying home from work/school: OR 0.78 (0.61, 1.00)

Mask use in public: OR 0.97 (0.68, 1.40)

Physical distancing in public: OR 0.93 (0.72, 1.19)

Positive antibody test results (n=47)

Social event avoidance: OR 0. 0.83 (0.19, 3.67)

Staying home from work/school: OR 1.39 (0.40, 4.84)

Physical distancing in public: OR 0.63 (0.19, 2.05)

We have decided not to include these results in our sensitivity analysis section, but would be open to doing so if the reviewer and/or editor feels strongly in favor. The reasons we used to decide against inclusion are:

1. Not meeting the proportional odds assumption across all subgroups

2. We prefer to maintain consistency in showing risk ratio and risk difference outcomes as opposed to the odds ratios produced by the proportional odds models

3. Risk ratios are more interpretable effect estimates than odds ratios and are appropriate for our longitudinal data structure. They also better handle data with very common outcomes, as ours are. Odds ratios produce inflated estimates relative to RRs in these situations. 

Thanks again for pointing us to this modeling strategy.

3. Furthermore, is there any specific reason not to include all data in a single analysis with two factors (seropositivity status and intervention) and their interaction effect? The imbalance might mitigate against this, but it would still be of interest to evaluate the interaction term.

Response: Thanks for the suggestion, and we agree with the reviewer that the study results could be strengthened by including a formal test of interaction between serostatus and intervention arm. We have now run the models as suggested, and include the Wald p-values for this interaction as footnotes to Table 2. Importantly, we observed null effects at 2-weeks for all behavioral outcomes and there was no evidence for statistical difference in effect by serostatus. The text of the footnote added to Table 2 reads as follows:

“Wald p-values for the interaction between serostatus and intervention arm were: 0.60 (social event avoidance), 0.27 (staying home from work/school), 0.08 (mask use in public), and 0.15 (physical distancing in public)”

4. And, along those lines, although it is probably only feasible (if at all) with the dichotomized version of the data (or with the Likert scale data treated as interval scaled), a full model would consider three fixed effects (seropositivity status, intervention, and time) and their interactions. However, this model would require a method for handling the repeated measures across time for each subject. This could be carried out using the 'nlme' R package, most likely (Pinheiro and Bates, 2017). However, convergence is always an issue for these binomial responses. But, treating the responses as intervals scaled should meet with no issues. There are several R packages that extend this modeling in a Bayesian framework as well as other frameworks such as STAN (Carpenter et al, 2017) which could also be used to perform such modeling.

Response: We agree with the reviewer that it would be very interesting to understand the three-way interaction between serostatus, intervention arm, and time. Unfortunately, our data were not structured or distributed to be able to answer this question. Specifically, we only have two time points for the SARS-CoV-2 serology. Importantly, after the second round of serology, no further behavioral outcomes were assessed and results were immediately returned to participants. Thus, we only have a single timepoint for which the randomized intervention applied and for which the behavioral outcomes were assessed.

5. An interesting alternative approach to the sensitivity analysis could be carried out using the concept of specification curve analysis (Simonsohn, Simmons, and Nelson, 2020). In a nutshell, the idea is to specify all of the alternative paths that the analysis could have taken to assess the impact of the one path that was actually chosen. It would be quite feasible to implement this analysis using the 'specr' R package (Masur and Scharkow, 2020).

Response: We appreciate this suggestion and learned a lot in doing some research into specification curve analysis. 

We do think that this suggested analysis would not have much added value with our particular dataset, however. In thinking through the ‘world’ of potential model specifications we could test as related to the outcome, we found that we do not have many other reasonable options to test. Cutoffs at different Likert values in behavioral outcome responses are not indicated due to sparse data in some of the response options. The alternate model specifications we were able to run [OLS and proportional odds models (see response to comment R3 above)] both produced results aligned with the null results we observed in our primary dichotomized analysis. 

The robustness of our findings across these specified models seems to suggest that there would be little added value from a specification curve analysis.

6. The study sample represents a subset of students who self-selected to join the study (34.4% of randomly selected, eligible IU-B undergraduate students). It could be that in this subset there is no observed risk compensation, but that in the complementary subset of students that did not self-select into the study there is some observed risk compensation. To be able to widen the scope of inference likely requires some assumptions about the comparability of these two subsets. There seems to be no real way to avoid this so it must be addressed in the limitations.

Response: This is a great point. We had a relatively low response rate from our randomly sampled study population (34.4%), leading to concerns that they may not represent the target population of all Indiana University undergraduate students. 

In general, the demographic distribution among our enrolled study participants does provide some reassurance on this point. In most respects, our study population reflected the demographic makeup of the underlying student body: by age, racial/ethnic background, on- vs. off-campus residence, and Greek affiliation. However, undergraduate women were overrepresented in our study population (64%) relative to the broader IU undergraduate population (50%). This gender difference would be important if women responded to antibody test information differently than men. 

More generally, to assume that our study population stands in for our larger target population, we have to assume that the enrolled study population has similar risk compensation responses as the people who did not enroll. We think that this assumption is generally reasonable, but acknowledge that we are not able to empirically test it.

We hope these edits to the 4th paragraph of the discussion provide a more transparent/thoughtful discussion of these concerns:

“We used random sampling from the IU student population and enrolled a large sample into the randomized trial. Though we observed a relatively low response rate, it is comparable to other response rates in university settings [28]. Nevertheless, to assume that our study population stands in for our larger target population, we have to assume that the enrolled study population has similar risk compensation responses as those who did not enroll. Undergraduate women, in particular, were overrepresented in our study population relative to the student body, Other demographic variables, however, generally tracked with those observed in the student body.”

7. The authors state "However, the results are likely to be generalizable to other young adult populations at other predominantly white universities similar to the one where this study was conducted." (lines 339-340). Is race a deciding factor here? If so, please provide some citations that would support this.

Response: We agree with the reviewer that the sentence as written may have been overly specific. We have now edited the sentence to read: 

“However, the results are likely to be generalizable to other young adult populations at other large, public universities similar to the one where this study was conducted.”

Thanks for the suggestion.

8. Not only did states differ in their approaches to handling COVID-19 issues broadly, but even cities within states differed, as did universities within states. And, students at different universities could come from a more or less heterogeneous background. It seems as though all of these effects make it more difficult to generalize rather than more easy.

Response: We agree with the reviewer that the generalizability of our findings depends at least partially on the COVID-19 policy environment. In addition to the edits we incorporated to the 4th paragraph of the discussion as outlined in response to R2, comment 6 above, the first two sentences of the paragraph now read:

“Results from this study likely do not generalize to university settings with different COVID-19 policies in place. However, the results are likely to be generalizable to other young adult populations at other large, public universities similar to the one where this study was conducted.”

9. The authors mention "potential for confounding" in line 347. Please list these potential issues, if only briefly or at a high level.

Response: Thanks for the opportunity to clarify this! We have now edited the first part of the 5th paragraph of the Discussion to more clearly lay out the potential for confounding were this study to be conducted observationally:

“Inference from this study is strengthened by the rigorous randomized study design used. By randomly assigning half of our participants to receive their results immediately, we were able to isolate the effect of receiving the results and minimize the potential for confounding to bias our results. Using a randomized trial design is critical because seeking testing for SARS-CoV-2, either for acute infection or antibody testing, requires both personal motivation and access to services which render comparisons between those who received testing and those who did not subject to significant possible bias. These factors, and those related to them, would be major sources of confounding in an observational study comparing those who did and did not receive antibody tests.”

10. In Figure 2, please show all available data rather than only three time points. Also, the bars in these figures appear offset in a way that makes them hard to interpret.

Although the stacked bar approach is not ideal, it is hard to suggest an alternative. One might attempt grouping bar charts but this is not likely to end well.

Response: Thanks for the suggestions for strengthening Figure 2. We have now added all follow-up timepoints to the graphs. 

We have also tried to improve the interpretability/clarity of the graphs by removing the gridlines and ensuring the axes are consistent for each subgroup within each behavioral outcome reported. We hope this is helps, and agree with the reviewer that the alternative (stacked bar charts) would not be as effective at conveying the information.

11. Given the linear model approach that was executed in the appendix, please also provide a plot of the least squares estimates over time for each combination of seropositivity status and intervention.

Response: We now include a supplemental figure (S2_Fig) that shows the linear model sensitivity analysis results over time – thanks for the suggestion. Although we agree with the reviewer that the added timepoints provide additional context for readers to absorb, we would like to emphasize that our study design necessitated the behavioral outcomes be interpreted at 2 weeks follow-up only. This is because our delayed arm participants received their test results at 4 weeks of follow up. So, at the 4-week timepoint and all subsequent timepoints, there was no exposure difference between arms (both arms had received their test results). 

R3 Comments

1. Title: it can be improved, highlight the object of the study.

Response: Thank you. We have now edited the title to read: 

“Does receiving a SARS-CoV-2 antibody test result change COVID-19 protective behaviors? Testing risk compensation in undergraduate students with a randomized controlled trial”

2. Abstract. I encourage the authors to add more detail about their core contributions in the abstract.

Response: Thanks for the opportunity to explain more clearly our core contribution. We’ve changed the Abstract text accordingly:

“Risk compensation, or matching behavior to a perceived level of acceptable risk, can blunt the effectiveness of public health interventions. One area of possible risk compensation during the SARS-CoV-2 pandemic is antibody testing. While antibody tests are imperfect measures of immunity, results may influence risk perception and individual preventive actions. We conducted a randomized control trial to assess whether receiving antibody test results changed SARS-CoV-2 protective behaviors.”

3. Introduction: The authors should make clearer what is the gap in the literature that is filled with this study. The authors must better frame their study within the vast body of literature that also addressed the issue of knowledge concerning COVID-19 that can affect the implementation of control measures in different groups of population (refer to articles with DOI: https://doi.org/10.3390/ijerph182010872).

Response: Please see the response to (2) above in clarifying the gap in the literature on the topic of risk compensation. We have also edited text in the 4th paragraph of the Introduction to make our contributions more explicit:

“Given the many influences on SARS-CoV-2 protective behaviors, a randomized trial is the most appropriate design to avoid confounding by these factors. Understanding the impact of information about past SARS-CoV-2 infection on preventive behavior is essential to managing viral control and for learning more about expected behavior post natural infection and vaccination. The SARS-CoV-2 vaccine has an overwhelmingly beneficial effect on lowering risk of serious COVID-19 disease and mortality [17]. However, understanding how much, if any, risk compensation might occur after natural infection or vaccination has important potential consequences for disease control.”

We agree that knowledge about COVID-19 is an important contributor to people’s uptake of preventive behavior and have edited and added citations in the third paragraph of the Introduction as follows:

“Learning the results of a SARS-CoV-2 test may influence individual behavior through several plausible mechanisms. Theoretically, risk compensation postulates that individuals have some amount of risk that they are willing to assume and will change their behaviors to match that level of risk [6]. Relatedly, behavioral disinhibition theory posits that feelings of protection against one health concern may cause people to engage in behaviors that put them at risk for other health issues. A necessary condition for these behavioral pathways to operate with SARS-CoV-2 tests is that people must have working perceptions that a relationship between prior infection and protection against future infections exists. These perceptions are likely related to overall COVID-19 and immunological knowledge.[7-9]”

4. Methods: The survey was conducted using a non-standard tool. The use of an unreliable instrument is a serious and irreversible limitation. A validation process must be performed to evaluate the tool. What about reliability, intelligibility and validation index? Was a pilot study performed?

Response: We used a portion of the World Health Organization’s survey tool for “Monitoring knowledge, risk perceptions, preventive behaviours and trust to inform pandemic outbreak response” to assess the outcomes of protective behaviors and have changed the text to report the origin of the survey questions. The conduct of a pilot study, while important for tool validation in non-urgent settings, would have precluded timely conduct of the study. The text we added to the 6th paragraph of the methods section is reproduced here:

“…participants were followed-up in parallel every two weeks with online surveys assessing their engagement in key COVID-19 protective behaviors using a scale from the World Health Organization COVID-19 survey tool.”

Citation: 

World Health Organization. Monitoring knowledge, risk perceptions, preventive behaviours and trust to inform pandemic outbreak response: Survey tool and guidance. 2020.

5. The enrolment procedure must be specified. How did the authors choose the way to select the sample? This can represent a great bias origin. How did they avoid the selection bias? The authors do not propose a minimum sample size. Without the numerical identification of the reference population is not clear the validity of the study. A non-representative sample is by its self a non-sense-survey.

Response: We clarified that we conducted a simple random sample of IU undergraduate students and include here the edited text and surrounding information that explains our sampling procedure. 

“We conducted a simple randomly sampled of all IU Bloomington undergraduate students enrolled at the beginning of the Fall 2020 semester yielding 7,499 students from the sampling frame of all IU Bloomington undergraduate students enrolled at the beginning of the Fall 2020 semester. Students in the sample were eligible to participate if they were (1) aged 18 years or older, (2) a current IU Bloomington undergraduate student, and (3) currently residing in Monroe County, Indiana. Of those sampled, 4,069 potential participants met the inclusion criteria for the study. All sampled students were contacted by email with a study invitation and a link to detailed information about study objective and procedures. The study team offered potential participants the opportunity for email or telephone consultations to answer any additional questions about study participation. After reviewing study information, interested and eligible students provided written informed consent remotely with an electronic signature.”

We take the concerns about the representativeness of our sample seriously and have made edits to the 4th paragraph of our discussion to more thoroughly and transparently address the potential threats to external validity. Please see our response to R2, comment 6 above for more details. 

6. Statistical analysis: I suggest to insert a measure of the magnitude of the effect for the comparisons. Please consider to include effect sizes.

Response: We apologize if we are misunderstanding the reviewer’s suggestion here, but we include both relative risks (RRs) and risk differences (RDs) as measures of effect for our primary and secondary outcomes in Table 2. We discuss these effect estimates in terms of magnitude (point estimates) and precision (95% CIs) in the table and text. Please let us know if there was something else you were suggesting.

7. Discussion: I also suggest expanding. Emphasize the contribution of the study to the literature. The discussion must be updated with the discussion regarding knowledge about the diseases (see the above mentioned reference) that can be an important confounding factors for this study. The Authors should add more practical recommendations for the reader, based on their findings. Also, the section of limitations and future search is also very short, the Authors could elaborate on that.

Response: Thanks for these suggestions to improve the discussion!

We place our findings in the current literature in the second paragraph of the discussion:

“That we found that behavior remained largely unchanged after receipt of antibody test results aligns with findings from prior studies of risk compensation in other settings. . Four interventions are often recognized for their potential to cause risk compensation: bicycle helmets, seatbelts, voluntary medical male circumcision and pre-exposure prophylaxis for HIV prevention, and HPV vaccination. Exhaustive reviews in each of these areas have consistently concluded that there is, in fact, little evidence for increased risk-taking after intervention exposure [10-13, 16, 28]. In spite of the weak record for risk compensation in other settings, many have raised concerns about the threat of risk compensation in the COVID-19 pandemic [29]. Vaccines and face mask mandates have been two areas of concern for risk compensation, but very few empirical studies have tested these associations. However, there is at least some suggestive evidence that risk compensation may, in fact, occur in settings with face mask mandates [30]. With the lack of risk compensation observed in our study, we provide additional empirical data to better shape our understanding of behavioral disinhibition in the COVID-19 pandemic.”

We now revisit the important point about COVID-19 knowledge and risk perception in the 3rd paragraph of the discussion:

“Relatedly, it is possible that risk behaviors did not differ between study arms because participants did not otherwise have strong perceptions of a relationship between serostatus and protection against future infections.”

We have added substantial discussion of study limitations in the 4th and 5th paragraphs, more thoroughly covering concerns about generalizability, loss to follow-up, confounding, and low seropositivity.

---

## [Decision Letter · Decision Letter 1]

6 Dec 2022

Does receiving a SARS-CoV-2 antibody test result change COVID-19 protective behaviors? Testing risk compensation in undergraduate students with a randomized controlled trial

PONE-D-22-18389R1

Dear Dr. Rosenberg,

We’re pleased to inform you that your manuscript has been judged scientifically suitable for publication and will be formally accepted for publication once it meets all outstanding technical requirements.

Kind regards,

Hermano Alexandre Lima Rocha

Academic Editor

PLOS ONE

Additional Editor Comments (optional):

Congratulations on your fine submission. We wish you all the best.

Reviewers' comments:

Reviewer's Responses to Questions

**Comments to the Author**

1. If the authors have adequately addressed your comments raised in a previous round of review and you feel that this manuscript is now acceptable for publication, you may indicate that here to bypass the “Comments to the Author” section, enter your conflict of interest statement in the “Confidential to Editor” section, and submit your "Accept" recommendation.

Reviewer #1: All comments have been addressed

Reviewer #2: All comments have been addressed

Reviewer #3: (No Response)

2. Is the manuscript technically sound, and do the data support the conclusions?

Reviewer #1: Yes

Reviewer #2: (No Response)

Reviewer #3: No

3. Has the statistical analysis been performed appropriately and rigorously? 

Reviewer #1: Yes

Reviewer #2: (No Response)

Reviewer #3: No

4. Have the authors made all data underlying the findings in their manuscript fully available?

Reviewer #1: Yes

Reviewer #2: (No Response)

Reviewer #3: No

5. Is the manuscript presented in an intelligible fashion and written in standard English?

Reviewer #1: Yes

Reviewer #2: (No Response)

Reviewer #3: Yes

6. Review Comments to the Author

Reviewer #1: No additional comments. On my opinion the information requested have been exhaustively addressed expecially for my concerns.

Reviewer #2: The authors have done an excellent job of thoroughly responding to all comments. The implementation of the analysis in SAS is nicely done as well.

Reviewer #3: The main problems of the questionnair validation and unreppresentative sample are still on. This make the paper no sense.

7. PLOS authors have the option to publish the peer review history of their article (what does this mean?). If published, this will include your full peer review and any attached files.

Reviewer #1: **Yes: **Manuela Monti

Reviewer #2: No

Reviewer #3: No

---

## [Editor Report · Acceptance letter]

12 Dec 2022

PONE-D-22-18389R1 

Does receiving a SARS-CoV-2 antibody test result change COVID-19 protective behaviors? Testing risk compensation in undergraduate students with a randomized controlled trial 

Dear Dr. Rosenberg:

I'm pleased to inform you that your manuscript has been deemed suitable for publication in PLOS ONE. Congratulations! Your manuscript is now with our production department. 

Kind regards, 

on behalf of

Dr. Hermano Alexandre Lima Rocha 

Academic Editor

PLOS ONE